# Human Body Restoration with
# One-Step Diffusion Model and A New Benchmark

**Jue Gong** [* 1]  **Jingkai Wang** [* 1]  **Zheng Chen** [1]  **Xing Liu** [2]  **Hong Gu** [2]  **Yulun Zhang** [† 1]  **Xiaokang Yang** [1]

## Abstract

Human body restoration, as a specific application of image restoration, is widely applied in practice and plays a vital role across diverse fields. However, thorough research remains difficult, particularly due to the lack of benchmark datasets. In this study, we propose a high-quality dataset automated cropping and filtering (HQ-ACF) pipeline. This pipeline leverages existing object detection datasets and other unlabeled images to automatically crop and filter high-quality human images. Using this pipeline, we constructed a person-based restoration with sophisticated objects and natural activities (*PERSONA*) dataset, which includes training, validation, and test sets. The dataset significantly surpasses other human-related datasets in both quality and content richness. Finally, we propose *OSDHuman*, a novel one-step diffusion model for human body restoration. Specifically, we propose a high-fidelity image embedder (HFIE) as the prompt generator to better guide the model with low-quality human image information, effectively avoiding misleading prompts. Experimental results show that OSDHuman outperforms existing methods in both visual quality and quantitative metrics. The dataset and code are available at: `https://github.com/gobunu/OSDHuman`.

## 1. Introduction

Human body restoration (HBR) aims to recover high-quality (HQ) images from low-quality (LQ) inputs featuring human figures. Unlike nature scene pictures, human figures naturally attract viewers' attention in images. However, real-world images often suffer from degradation during capture and transmission, such as blur, noise, resolution reduction, and JPEG artifacts. These distortions severely impact the

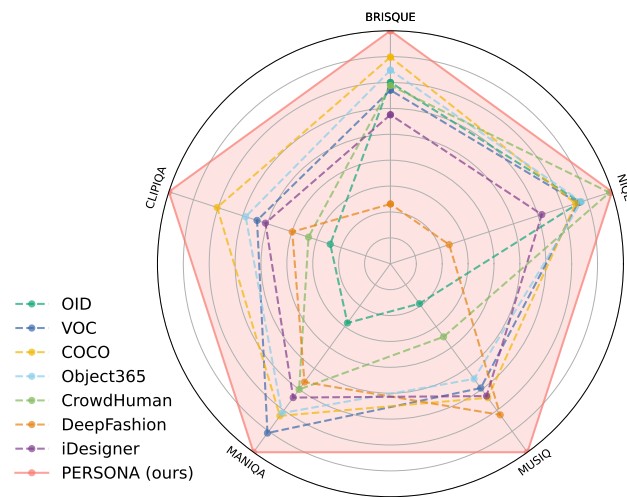

*Figure 1.* Comparison of no-reference image quality assessment metrics across human-related datasets. The object detection datasets specifically evaluate subsets with humans. Our proposed PERSONA dataset outperforms others significantly.

recognition of human activities and the extraction of information from the image. Furthermore, degraded images make other human-related downstream tasks more challenging, such as human-object interaction detection (Wang et al., 2024d; Liu et al., 2025), human pose estimation (Samkari et al., 2023; Atmosukarto et al., 2024), and 3D reconstruction (Wang et al., 2021a; Sun et al., 2024).

Despite its practical significance, progress in HBR remains constrained, primarily due to the absence of task-specific benchmark datasets. In natural scenarios, humans exhibit a wide range of activities and complex interactions with their surroundings. Therefore, the benchmark dataset for HBR must be large, cover complex scenarios, and include natural activities. Datasets in the fashion domain, such as DeepFashion (Liu et al., 2016) and iDesigner (Dufour et al., 2022), focus on runway or studio scenarios. As a result, these datasets contain only limited types of human actions, making them unsuitable for HBR. Moreover, human images often involve multiple individuals interacting with each other, adding further complexity to the restoration task. Such complexity makes datasets focused on single-person image generation, such as SHHQ (Fu et al., 2022) and CosmicMan-HQ (Li et al., 2024b), less suitable for HBR, as they mainly focus on single-person images.

---

[*]Equal contribution  [1]Shanghai Jiao Tong University, China [2]vivo Mobile Communication Co., Ltd, China. Correspondence to: [†]Yulun Zhang <yulun100@gmail.com>.

*Proceedings of the 42^{nd} International Conference on Machine Learning*, Vancouver, Canada. PMLR 267, 2025. Copyright 2025 by the author(s).

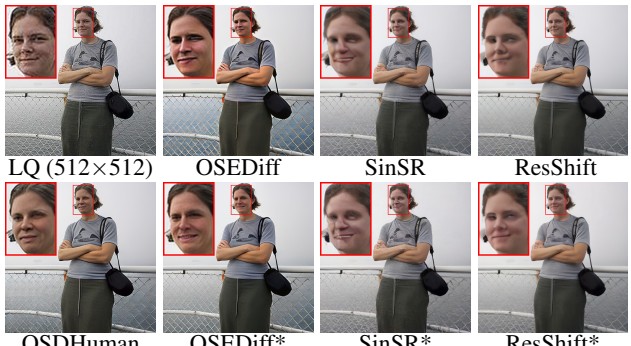

*Figure 2.* Visual examples of diffusion-based image restoration methods evaluated on PERSONA-test. The asterisk (*) indicates methods retrained on PERSONA dataset. Our OSDHuman produces more natural and faithful visual results compared to others.

Additionally, an HBR model can achieve optimal performance only when trained on sufficiently high-quality datasets. Degraded datasets could cause bias in the model's weights, as it is difficult to distinguish between degradation and features in LQ images. Some existing image restoration datasets, such as LSDIR (Li et al., 2023) and DIV2K (Agustsson & Timofte, 2017), are of high quality and cover complex real-world scenarios. However, the proportion of human images in these datasets is small, and they are not specifically tailored for human images. Other human-related high-level datasets, such as those for object detection (Kuznetsova et al., 2020; Shao et al., 2019) and keypoint detection (Lin et al., 2014), have a wider range of human activities and sophisticated surrounding objects due to the diversity of image sources. However, as illustrated in Fig. 1, these datasets lack dedicated quality filtering, containing substantial LQ samples.

However, even with a high-quality benchmark, achieving excellent HBR performance still requires a well-designed model architecture. In recent studies, image restoration models with latent diffusion model (LDM) (Rombach et al., 2022) architecture achieve promising results due to powerful generative capabilities. These models combine generation and restoration to reconstruct lost parts of LQ images using the features provided. They primarily use two main latent space mapping methods: variational autoencoder (VAE) (Kingma & Welling, 2014) and vector quantized VAE (VQVAE) (Oord et al., 2017). As shown in Fig. 2, models with VQVAE, such as ResShift (Yue et al., 2023) and SinSR (Wang et al., 2024e), often produce distorted structures in detail. Even with retraining, these issues are difficult to resolve due to the VQVAE codebook's inability to capture the fine details of the human body. On the other hand, models with VAE, like OSEDiff (Wu et al., 2024a), have better generalization and detail generation capabilities, but they are still not specifically optimized for HBR.

While multi-step diffusion models have strong restoration abilities for LQ images, they often require substantial com-

putational resources, which limits their applicability. To reduce resource consumption, one-step diffusion (OSD) models are proposed and achieve good results. By leveraging large-scale pretrained text-to-image (T2I) models (Saharia et al., 2022; Rombach et al., 2022) as foundation models, OSD models combine generation power with fast inference. Therefore, OSD models are highly competitive in HBR. However, this also requires the model to incorporate an appropriate prompt extractor that can derive a high-fidelity prompt from complex human images. Otherwise, the resulting prompt could mislead the restoration of the model.

To address the limitations, we propose OSDHuman, a novel OSD model for HBR. **Firstly**, to overcome the lack of benchmark datasets in HBR, we propose a high-quality dataset automated cropping and filtering (HQ-ACF) pipeline. This pipeline preprocesses both labeled and unlabeled datasets to isolate images containing humans. Then it refines human bounding boxes and crops them accordingly. Using no-reference image quality assessment (IQA) metrics, we ultimately produce a dataset. **Secondly**, leveraging HQ-ACF, we develop a person-based restoration with sophisticated objects and natural activities (PERSONA) dataset, which comprises 109,053 HQ 512×512 human images for training. This pipeline also provides images for validation and testing. The PERSONA dataset includes both individual-environment interactions and multi-person interactions, averaging 3.4154 individuals per image. **Thirdly**, to provide prompts suitable for HBR, we propose a high-fidelity image embedder (HFIE). HFIE uses an image encoder from RAM (Zhang et al., 2023) and a multi-head attention layer with a learnable embedding as the query. This design avoids distortions introduced by tags when summarizing images, thereby preventing misleading prompts that could impair model restoration. In addition, we employ a variational score distillation (VSD) regularizer to guide the model's generative distribution toward natural image distributions.

Our contributions can be summarized as follows.

- We propose a person-based restoration with sophisticated objects and natural activities (PERSONA) dataset which provides a benchmark for human body restoration, encompassing training, validation, and test sets.

- Our PERSONA dataset surpasses other human-related datasets in quality and includes a wide range of scenarios that cover the majority of human activities.

- We propose OSDHuman, an innovative one-step diffusion model for human body restoration. It features a high-fidelity image embedder (HFIE) designed for extracting suitable prompts from human images.

- Our OSDHuman achieves state-of-the-art human body restoration performance, excelling in visual quality and metrics while maintaining lower computational costs.

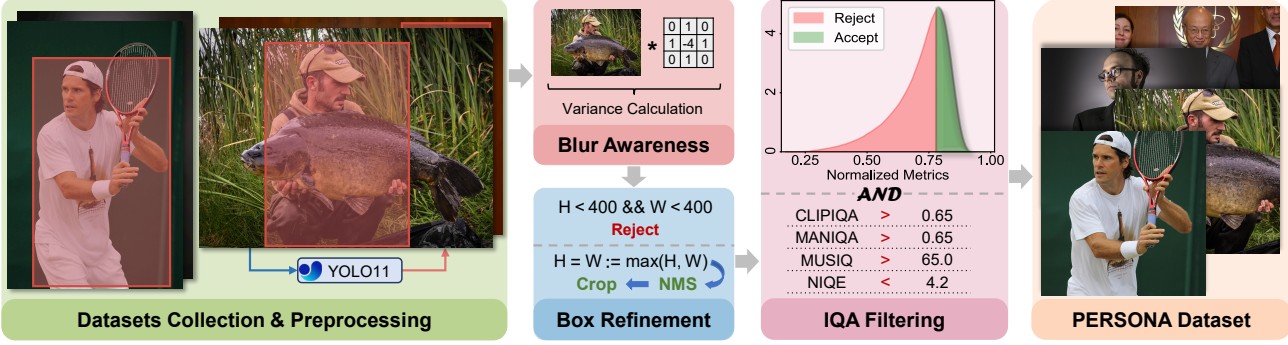

*Figure 3.* High-quality dataset automated cropping and filtering pipeline. The pipeline consists of four stages. **First**, multiple datasets are collected, comprising millions of images. Images without labels are processed using YOLO11 for human detection. **Then**, a Laplacian operator is applied to compute image Laplacian variance, filtering out images below a threshold. **Next**, human boxes are adjusted to the square shape, and overly small or densely packed boxes are removed. **Finally**, cropped human images are evaluated using Image Quality Assessment (IQA) metrics. Images ranking in the top third by normalized metrics and exceeding the metric threshold are selected. These 109,053 images constitute the person-based restoration with sophisticated objects and natural activities (PERSONA) dataset.

## 2. Related Works

### 2.1. Human Body Restoration

Human body restoration (HBR) could benefit both the fashion industry for display and the photography and camera producers. The purpose is evident, focusing on the human body and making it look better. Compared to general image restoration for arbitrary objects, HBR is more constrained, allowing for the use of a variety of prior knowledge. Firstly, current research on image segmentation has made significant progress in generating segmentation masks for different body parts, such as hair and arms, which provide a clear description of the human body shape. This knowledge is beneficial for handling the boundaries in the HBR tasks. Since the human body composition is relatively fixed and limited, it is easier to estimate compared to the random and arbitrary objects in natural images.

Lots of research has been done recently. A previous work (Liu et al., 2021a) captures body texture using sub-bands of the non-subsampled shearlet transform, while PRCN (Wang et al., 2024c) employs a pyramid residual network to estimate texture and shape priors, enhancing body images. DiffBody (Zhang et al., 2024), as the first to apply diffusion models, uses pose-attention, text guidance, and a body-centered sampler to integrate semantic information for body-region enhancement.

### 2.2. Diffusion Models

Since the diffusion model was released and popular, many efforts have been made. Two classical applications are image restoration and text-to-image (T2I). Image restoration, as the first and most natural application, has been developed a lot (Saharia et al., 2023; Whang et al., 2022; Avrahami et al., 2022; Chen et al., 2023; Xia et al., 2023). With the development of conditional diffusion models (Rombach et al., 2022), numerous companies have invested heavily in training more powerful T2I models. Recently, many efforts have been made to integrate these two typical applications. The rapidly developing Stable Diffusion (Rombach et al., 2022), DALLE (Ramesh et al., 2021), and PixArt (Chen et al., 2024) have continually pushed the boundaries of realism and diversity in T2I generation. Many image restoration methods have also leveraged pretrained models to achieve more natural image recovery (Wu et al., 2024b; Yang et al., 2024; Lin et al., 2024; Wu et al., 2024a; Wang et al., 2024a).

Stricted on the multi-step in the diffusion inference procedure, the diffusion models with 50 or more steps (Wang et al., 2024b; Lin et al., 2024; Wu et al., 2024b; Yang et al., 2024) cannot actually be used in practice. Many efforts have been made to faster diffusion models, such as cutting, quantizing, and compressing. Moreover, eliminating the number of inference timesteps is a convincing way, especially applied in image restoration. SinSR (Wang et al., 2024e) pioneers one-step inference for diffusion-based super-resolution (SR) by distilling deterministic generation functions into a student network, coupled with a consistency-preserving loss and efficient training pair generation strategy. OSEDiff (Wu et al., 2024a) adapts pretrained SD models for SR through LoRA-finetuned U-Net and variational score distillation, enabling direct low-quality image reconstruction in one step without noise injection. Those methods achieve a fascinating performance in natural image restoration.

## 3. Methods

### 3.1. High Quality Human Dataset Pipeline

For image restoration tasks, large-scale high-quality datasets are required to simulate various real-world scenarios and objects. There are already many high-quality image restoration datasets, such as FFHQ (Karras et al., 2019) and LSDIR (Li et al., 2023). However, these datasets are not ideally suited for the human body restoration (HBR) task due to their lack

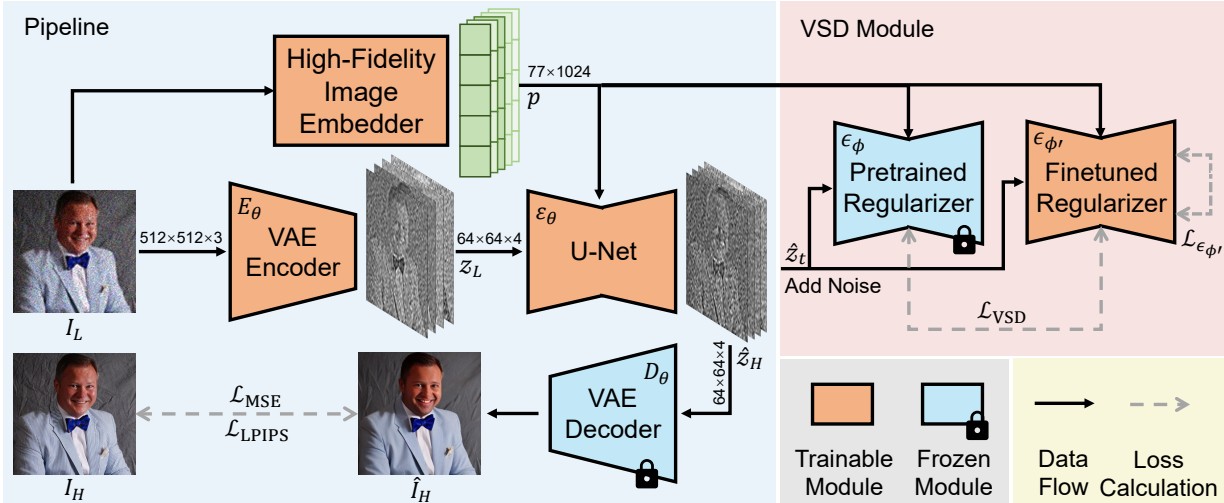

*Figure 4.* Training Framework of OSDHuman. **First**, the LQ image $I_L$ is processed through the VAE Encoder, U-Net, and VAE Decoder, ultimately producing the restored HQ image $\hat{I}_H$. The conditional input of the U-Net is provided by the high-fidelity image embedder (HFIE). **Second**, during the training process, the $\hat{z}_H$ generated by the U-Net is subjected to noise and then passed through the pretrained and finetuned regularizers. $\mathcal{L}_{VSD}$ represents the distribution's difference between the model output and the natural image. $\mathcal{L}_{VSD}$, together with $\mathcal{L}_{LPIPS}$ and $\mathcal{L}_{MSE}$, constitutes the training objective. **In summary**, during the training stage, the VAE Encoder, U-Net, and finetuned regularizer are trained with LoRA, while other modules remain frozen. During inference, the VSD module is not utilized.

of focus on human-specific features. Moreover, the dataset should enable models to adapt to real-world environments' complexities. It must encompass most scenarios including interactions among people and between people and their surroundings. Therefore, we propose a high-quality dataset automated cropping and filtering (HQ-ACF) pipeline for HBR datasets, as well as a person-based restoration with sophisticated objects and natural activities (PERSONA) dataset.

**Automated Cropping and Filtering Pipeline.** As illustrated in Fig. 3, we first collect a series of commonly used and publicly available large-scale object detection datasets, including COCO (Lin et al., 2014), OID (Kuznetsova et al., 2020; Krasin et al., 2017), Object365 (Shao et al., 2019) and CrowdHuman (Shao et al., 2018), comprising approximately 4 million images. We then filter the images by labels, selecting those containing "human" or synonymous labels, such as "Human Body" in OID and "person" in Object365. To further refine the selection, we conduct human detection on the image restoration dataset LSDIR (Li et al., 2023), using the YOLO11 model (Jocher & Qiu, 2024) for processing. This operation resulted in bounding boxes similar to those in object detection datasets.

Next, we apply the Laplacian operation to these datasets and compute the variance of the results. Images with a variance below the threshold are discarded, as these correspond to images with a high degree of blurriness. Before cropping, we also check the size of the human bounding boxes. Images with low-resolution human bodies are rejected. After these steps, we use the bounding boxes to crop the images. When cropping, the side length of the bounding box's longer edge is used as the side length of the cropping box, ensuring a

square crop. In cases where an image contains multiple overlapping boxes, non-maximum suppression (NMS) is applied, prioritizing the box closest to the image center. The cropped images are then resized to $512 \times 512$.

Finally, we obtain approximately 440,000 cropped images, on which we measure no-reference Image Quality Assessment (IQA) metrics. The metrics used are common in image restoration, including CLIPIQA (Wang et al., 2023a), MANIQA (Yang et al., 2022), MUSIQ (Ke et al., 2021), and NIQE (Zhang et al., 2015). To balance the evaluation performance of each metric, we normalize the obtained IQA metrics using the following standardization formula:

$$\text{Normalized Metrics} = \frac{1}{N} \sum_{i=1}^{N} \frac{M_i - \mu_i}{\sigma_i}, \quad (1)$$

where $\mu_i$ is the mean and $\sigma_i$ is the standard deviation of the metrics ($M_i$). Since NIQE is better when its score is smaller, we first apply a negative transformation to its values. The normalized scores are then accumulated per image and sorted. The final PERSONA dataset version consists of 109,053 images with normalized metrics in the top third, and each IQA metric must exceed a predefined threshold.

### 3.2. One-Step Diffusion (OSD) Model

**Model Architecture Overview.** Most image restoration tasks with OSD models are extensively studied in previous works (Wang et al., 2024e; Wu et al., 2024a; Wang et al., 2024a; Li et al., 2024a). However, these methods struggle to achieve desirable results in human body restoration (HBR). To address this limitation, we propose OSDHuman, an OSD model specifically designed for HBR. Specifically, we adopt

a Stable Diffusion (SD) model architecture (Rombach et al., 2022) by fixing the number of steps, thereby transforming it into an OSD framework. As shown in Fig. 4, the first step uses the variational autoencoder (VAE) encoder $E_\theta$ to project the low-quality (LQ) image $I_L$ into the latent space, resulting in $z_L = E_\theta(I_L)$. Subsequently, a single denoising operation $F_\theta$ is applied to estimate the noise, which is crucial for enabling the calculation of the predicted high-quality (HQ) latent vector $\hat{z}_H$ through the equation:

$$\hat{z}_H = F_\theta(z_L; p) = \frac{z_L - \sqrt{1 - \bar{\alpha}_{T_L}} \varepsilon_\theta(z_L; p, T_L)}{\sqrt{\bar{\alpha}_{T_L}}}, \quad (2)$$

where $\varepsilon_\theta$ represents the denoising network governed by the parameter $\theta$, $p$ is the output of the high-fidelity image embedder (HFIE), and $T_L$ refers to the diffusion timestep. A predefined parameter $T_L \in [0, T]$ is used as input to the U-Net, where $T$ signifies the total number of diffusion steps (e.g., $T = 1,000$ in SD). The VAE decoder $D_\theta$ is then employed to reconstruct the HQ image $\hat{I}_H$ from the predicted latent vector $\hat{z}_H$, expressed as $\hat{I}_H = D_\theta(\hat{z}_H)$. If the generator is denoted as $\mathcal{G}$, the complete process can be summarized by the following equation:

$$\hat{I}_H = \mathcal{G}_\theta(I_L; p). \quad (3)$$

**Training Objective.** During training, we utilize pixel-wise MSE loss and perceptual loss LPIPS (Zhang et al., 2018). Additionally, the obtained $\hat{z}_H$ is used to compute the variational score distillation (VSD) loss, ensuring alignment between the generated images and natural images. The final overall training objective for the generator $\mathcal{G}_\theta$ is the following. $\lambda_1$ and $\lambda_2$ are the weights for $\mathcal{L}_{\text{LPIPS}}$ and $\mathcal{L}_{\text{VSD}}$.

$$\begin{aligned} \mathcal{L}_{\mathcal{G}_\theta} = \mathcal{L}_{\text{MSE}}(I_H, \hat{I}_H) &+ \lambda_1 \cdot \mathcal{L}_{\text{LPIPS}}(I_H, \hat{I}_H) \\ &+ \lambda_2 \cdot \mathcal{L}_{\text{VSD}}(\hat{z}_H, p). \end{aligned} \quad (4)$$

**High-Fidelity Image Embedder.** The degradation level of LQ human images is generally significant, and the image content is often highly complex. It is necessary to consider the coordination between the human pose and the surrendering. Therefore, we propose an HFIE that can guide the restoration direction, reducing the feature gap between HQ and LQ human images. OSEDiff (Wu et al., 2024a) employs a finetuned RAM (Zhang et al., 2023) as the degradation-aware prompt extractor (DAPE). It provides tags to guide the OSD model. However, the tags generated by DAPE are often too broad and imprecise for human images, offering insufficient and even biased guidance to the OSD model.

As shown in Fig. 5, the RAM used in DAPE consists of two parts: the image encoder and the tagging head. However, HFIE only uses the image encoder ($\mathcal{E}$) in RAM, leveraging the Swin Transformer (Liu et al., 2021b), which downsamples the input LQ $I'_L \in \mathbb{R}^{384 \times 384 \times 3}$ by a factor of 32. The $I'_L$ represents the resized LQ $I_L$. The image embeddings are $x_L = \{x_{L,k} \in \mathbb{R}^{512}\}_{k=1}^{145}$, where the first 144 embeddings

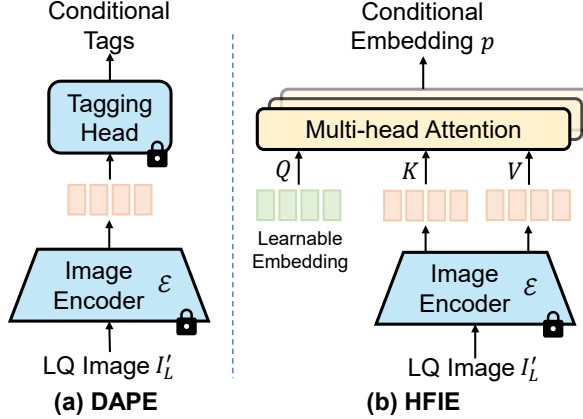

*Figure 5.* Comparison of the architectures of HFIE and DAPE.

represent local information of images. The remaining 1 embedding is obtained through the average pooling of others, which contains the overall information of images. Therefore, using a linear layer to reduce the embeddings' size to match the input of SD (e.g., $77 \times 1,024$ for SD-2.1) would result in the loss of distinguish ability between overall and local information. Our proposed method HFIE uses learnable embeddings $Q$ as the query input to the multi-head attention (MHA) layer, ultimately producing the HFIE:

$$p = \text{HFIE}(I'_L) = \text{MHA}(Q, \mathcal{E}(I'_L), \mathcal{E}(I'_L)). \quad (5)$$

**Variational Score Distillation (VSD).** Fine-tuning image restoration models often encounter challenges due to the limited training data available, especially when compared to large-scale foundation models like Stable Diffusion (SD). It leads to generated images that fail to align with the natural image space distribution. Previous studies (Yin et al., 2023; Wang et al., 2023b; Dao et al., 2024) propose some VSD methods to solve this problem by aligning the distributions represented by two diffusion models. Following OSED-iff (Wu et al., 2024a), we apply a VSD module in latent space to guide OSDHuman in learning the distribution of natural images from SD. The VSD loss is computed from the distribution gap in the latent space output by the pre-trained regularizer $\epsilon_\phi$ and finetuned regularizer $\epsilon_{\phi'}$. The gradient of the VSD loss is defined as:

$$\begin{aligned} \nabla_\theta \mathcal{L}_{\text{VSD}}(\hat{z}_H, p) &= \nabla_{\hat{z}_H} \mathcal{L}_{\text{VSD}}(\hat{z}_H, p) \frac{\partial \hat{z}_H}{\partial \theta} \\ &= \mathop{\mathbb{E}}_{t, \epsilon, \hat{z}_t} \left[ \frac{\epsilon_\phi(\hat{z}_t; t, p) - \epsilon_{\phi'}(\hat{z}_t; t, p)}{\text{mean}(||\epsilon_\phi(\hat{z}_t; t, p) - \hat{z}_H||)} \cdot \frac{\partial \hat{z}_H}{\partial \theta} \right], \end{aligned} \quad (6)$$

where $t$ is sampled from the range $[20, 980]$, $\varepsilon \sim \mathcal{N}(0, I)$ and $\hat{z}_t$ denotes the output after adding noise at timestep $t$. Besides, to ensure VSD working, the finetuned regularizer $\epsilon_{\phi'}$ needs to be trainable. Its training objective is:

$$\mathcal{L}_{\epsilon_{\phi'}} = \mathop{\mathbb{E}}_{t, \epsilon, p, \hat{z}_H} \mathcal{L}_{\text{MSE}}\left(\epsilon_{\phi'}(\hat{z}_t; t, p), \epsilon\right). \quad (7)$$

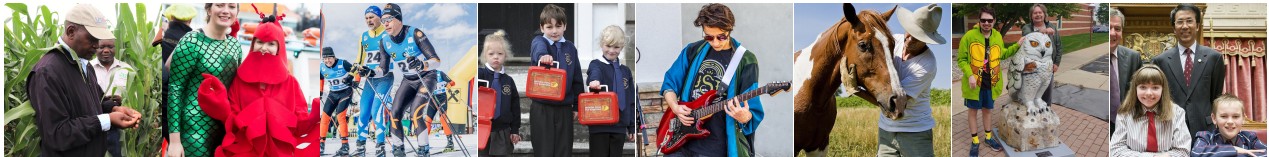

*Figure 6.* The PERSONA dataset consists of human images engaged in various natural activities, featuring diverse surrounding objects.

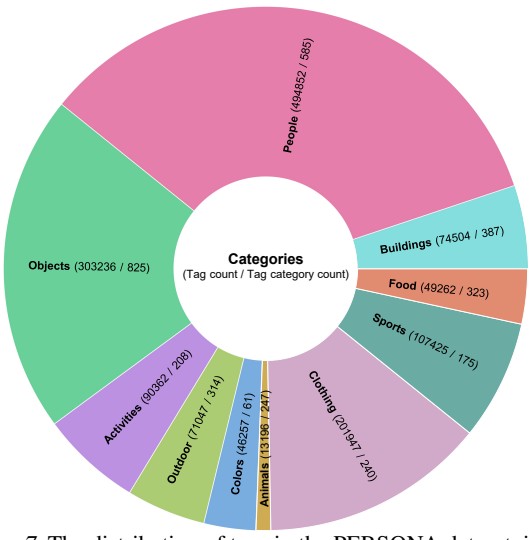

*Figure 7.* The distribution of tags in the PERSONA dataset, identified by the Recognize Anything Plus Model (Huang et al., 2023). The angles of the pie chart represent the frequency of each tag category in the dataset, i.e., the tag count. The tag category count indicates how many tags are contained within each category.

## 4. Experiments

### 4.1. Quantitative Analysis of PERSONA Dataset

**High-Quality Dataset**. Our person-based restoration dataset with sophisticated objects and natural activities (PERSONA) dataset, consists of 109,053 human images with a resolution of 512×512. These images are obtained through the high-quality automated cropping and filtering (HQ-ACF) pipeline. To quantify the quality of the PERSONA dataset, we evaluated a range of no-reference image quality assessment (IQA) scores, including CLIP-IQA (Wang et al., 2023a), MANIQA (Yang et al., 2022), MUSIQ (Ke et al., 2021), BRISQUE (Mittal et al., 2011), and NIQE (Zhang et al., 2015). We compare the results with those from other human-related datasets, including the object detection datasets OID (Kuznetsova et al., 2020; Krasin et al., 2017), VOC (Everingham et al., 2010), COCO (Lin et al., 2014), Object365 (Shao et al., 2019), CrowdHuman (Shao et al., 2018), and fashion domain datasets DeepFashion (Liu et al., 2016), iDesigner (Dufour et al., 2022). As shown in Tab. 1, our dataset consistently outperforms these datasets across all the no-reference IQA measures.

**Rich-Diversity Dataset.** We utilize the Recognize Anything Plus Model (Huang et al., 2023) to obtain image understanding tags for our PERSONA dataset. The visual results are shown in Fig. 7. **In total**, 3,365 distinct tag

| Dataset | BRISQUE↓ | NIQE↓ | CLIPIQA↑ | MANIQA↑ | MUSIQ↑ |
|---|---|---|---|---|---|
| OID | 19.8621 | 3.6611 | 0.4775 | 0.5940 | 60.4947 |
| VOC | 21.2764 | 3.7155 | 0.6071 | 0.7011 | 68.6073 |
| COCO | 15.2091 | 3.7774 | 0.6778 | 0.6844 | 69.5428 |
| Object365 | 17.6128 | 3.6315 | 0.6273 | 0.6817 | 67.7270 |
| CrowdHuman | 20.4306 | 2.9283 | 0.5160 | 0.6587 | 63.6830 |
| DeepFashion | 42.1884 | 6.9403 | 0.5448 | 0.6515 | 71.1873 |
| iDesigner | 25.8027 | 4.6227 | 0.5922 | 0.6666 | 69.3768 |
| **PERSONA** (ours) | **10.3758** | **2.8659** | **0.7632** | **0.7198** | **74.7808** |

*Table 1.* Quantitative comparison across different human-related datasets, with the best results highlighted in **bold**.

categories are identified from the dataset. And the largest proportion belongs to the "Objects" category, demonstrating the presence of sophisticated objects in our dataset. **Additionally**, the dataset generates a total of 1,452,088 tags, with approximately half of these tags falling under the "People", "Sports", and "Activities" categories. It indicates that the dataset is centered on natural human activities. As shown in Fig. 6, most of the images relate to this theme. **Finally**, the average number of tags per image in the dataset is 13.32, highlighting the dataset's ease of understanding.

### 4.2. Experimental Settings

**Training and Testing dataset.** Our OSDHuman model is trained on our PERSONA dataset, which contains 109,053 high-quality 512×512 human images. The degradation pipeline of Real-ESRGAN (Wang et al., 2021b) is used to generate synthetic degraded images for training. The test data includes PERSONA-Val and PERSONA-Test, both generated by our HQ-ACF pipeline. The HQ images in the validation set are specially selected from those that comply with the pipeline, ensuring that no images in the validation set share sources with the training set. A total of 4,216 images are used, and the degraded LQ images are generated using the same degradation pipeline as during training. The test set is derived from the VOC dataset (Everingham et al., 2010) by performing a partial crop using the HQ-ACF pipeline, followed by sampling under predefined IQA thresholds, yielding 3,000 images with real-world LQ.

**Evaluation Metrics.** For the *PERSONA-Val*, we employ both reference-based and non-reference IQA metrics. DISTS (Ding et al., 2020) and LPIPS (Zhang et al., 2018) are used for reference-based perceptual quality assessment, while PSNR and SSIM (Wang et al., 2004) (calculated on the Y channel in YCbCr space) are used for reference-based fidelity assessment. Additionally, FID (Heusel et al., 2017) is used to measure the distribution between the restored images and GT. The non-reference IQA metrics we used includes CLIPIQA (Wang et al., 2023a), MUSIQ (Ke et al.,

| Type | Methods | PERSONA-Val | | | | | | | | | PERSONA-Test | | | |
|---|---|---|---|---|---|---|---|---|---|---|---|---|---|---|
| | | DISTS↓ | LPIPS↓ | PSNR↑ | SSIM↑ | FID↓ | CLIPIQA↑ | MANIQA↑ | MUSIQ↑ | NIQE↓ | CLIPIQA↑ | MANIQA↑ | MUSIQ↑ | NIQE↓ |
| Multi-Step Diffusion | DiffBIR | 0.1475 | 0.3047 | 21.52 | 0.5718 | 14.8418 | 0.8080 | 0.7030 | 76.4751 | 3.9752 | 0.7287 | 0.6812 | 73.2505 | 4.9820 |
| | SeeSR | 0.1379 | 0.2851 | 21.31 | 0.5955 | 15.0063 | 0.7785 | 0.6993 | 76.8001 | 3.6125 | 0.6716 | 0.6698 | 73.2988 | 4.0228 |
| | PASD | 0.1891 | 0.3587 | 22.17 | 0.6154 | 26.7405 | 0.5950 | 0.6090 | 67.3329 | 4.6249 | 0.5765 | 0.6703 | 72.1972 | 3.8728 |
| | ResShift | 0.1795 | 0.3313 | 22.10 | 0.6157 | 30.7865 | 0.5931 | 0.5833 | 69.5889 | 4.7448 | 0.5544 | 0.6101 | 69.4611 | 4.8438 |
| | ResShift* | 0.1822 | 0.3372 | 21.73 | 0.5969 | 29.4177 | 0.6721 | 0.6121 | 71.9257 | 4.8061 | 0.6130 | 0.6174 | 70.2313 | 4.8735 |
| One-Step Diffusion | SinSR | 0.1691 | 0.3187 | 21.92 | 0.5967 | 22.9041 | 0.6372 | 0.5712 | 70.0839 | 4.4392 | 0.5882 | 0.6010 | 69.0157 | 4.7510 |
| | SinSR* | 0.1844 | 0.3348 | 21.54 | 0.5766 | 34.5773 | 0.7033 | 0.5819 | 71.2943 | 4.6294 | 0.6936 | 0.5962 | 69.9375 | 4.9873 |
| | OSEDiff | 0.1510 | 0.2824 | 21.81 | 0.6182 | 17.6308 | 0.6875 | 0.6639 | 74.0774 | 3.5858 | 0.6734 | 0.6919 | 73.5634 | 4.4600 |
| | OSEDiff* | 0.1476 | 0.2756 | 22.23 | 0.6342 | 17.2200 | 0.7034 | 0.6976 | 73.7636 | 3.9980 | 0.6874 | 0.7052 | 73.1611 | 4.4261 |
| | **OSDHuman** | 0.1414 | 0.2627 | 22.41 | 0.6363 | 16.5987 | 0.7295 | 0.6934 | 76.1256 | 3.5750 | 0.7155 | 0.6977 | 73.7694 | 4.1287 |

*Table 2.* Quantitative comparisons on synthetic PERSONA-Val and real-world PERSONA-Test datasets. For each metric, the best and second-best results are highlighted in red and cyan, within both multi-step and one-step diffusion-based methods. Models labeled with an asterisk (*) represent versions retrained on our PERSONA dataset for reference.

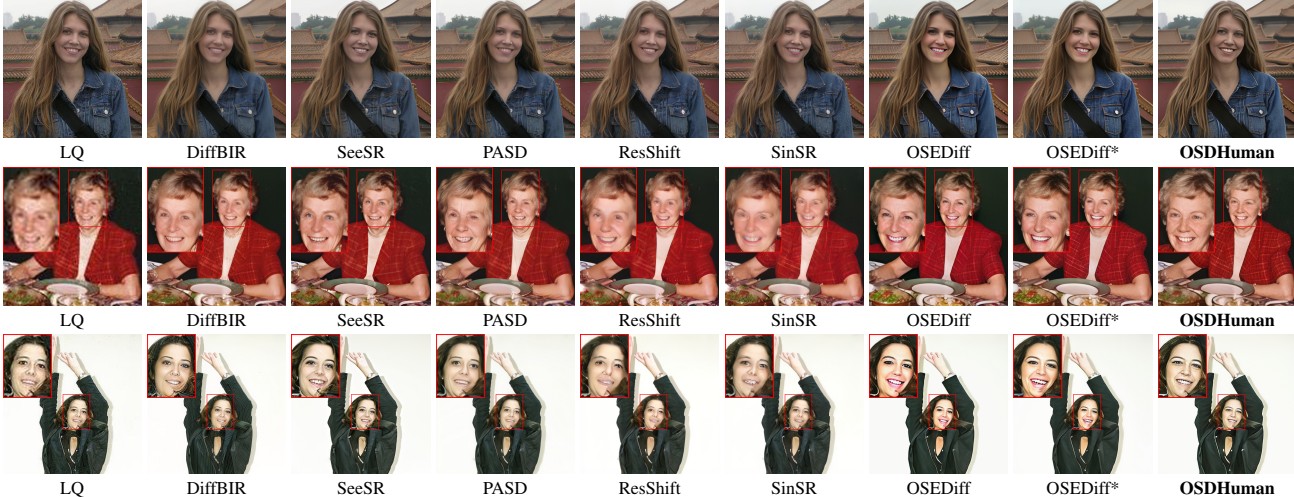

*Figure 8.* Visual comparison of the real-world PERSONA-Test dataset in challenging cases. Please zoom in for a better view.

2021), MANIQA (Yang et al., 2022), and NIQE (Zhang et al., 2015). For the *PERSONA-Test*, we use the same non-reference IQA metrics as PERSONA-Val. We utilize the evaluation codes provided by pyiqa (Chen & Mo, 2022), with the pipal version used for MANIQA.

**Implementation Details.** The OSDHuman model is trained by AdamW optimizer (Loshchilov & Hutter, 2019) with a batch size of 16 and 5e-5 learning rate. The Stable Diffusion v2-1 model (Stability AI, 2022) serves as the pretrained OSD model with the timestep frozen to 999, and the prompt embedding is provided by HFIE. The LoRA (Hu et al., 2022) rank for the VAE encoder, the U-Net of the generator, and the regularizer are all set to 4. The weighting scalars $\lambda_1$ and $\lambda_2$ in Eq. 4 are set to 2 and 1, respectively. Training is conducted for 35K iterations on 4 NVIDIA A800 GPUs.

**Compared Methods.** We compare OSDHuman with several diffusion-based methods, including DiffBIR (Lin et al., 2024), SeeSR (Wu et al., 2024b), PASD (Yang et al., 2024), ResShift (Yue et al., 2023), SinSR (Wang et al., 2024e) and OSEDiff (Wu et al., 2024a). Among them, SinSR and OSEDiff are OSD models. ResShift and OSEDiff are retrained on our PERSONA dataset, referred to as ResShift* and OSEDiff*, respectively. Additionally, SinSR is distilled using ResShift*, namely SinSR*.

### 4.3. Main Results

**Quantitative Comparisons.** Tab. 2 presents the evaluation metrics of our OSDHuman on the synthetic PERSONA-Val and real-world PERSONA-Test datasets. Our method achieves the best or second-best results across most metrics when compared with both original and retrained one-step diffusion models. Against multi-step diffusion methods, OS-DHuman outperforms in DISTS, LPIPS, PSNR, SSIM, and NIQE on PERSONA-Val, as well as CLIPIQA, MANIQA, and MUSIQ on PERSONA-Test, with other metrics being comparable. While multi-step diffusion methods excel in reconstructing highly degraded regions and achieving higher CLIPIQA scores, their generated content often deviates from the original, resulting in lower fidelity and perceptual scores such as PSNR, SSIM, DISTS, and LPIPS.

**Visual Comparisons.** As shown in Figs. 8 and 9, representative images from the real-world PERSONA-Test dataset and the synthetic PERSONA-Val dataset are visualized. Existing state-of-the-art image restoration methods are not well suited for human body restoration (HBR). In HBR tasks, the most challenging aspects often involve parts of the human image with intricate textures and delicate structures, such as faces, fingers, and surrounding objects. These methods frequently exhibit issues like over-smoothing or unnatural

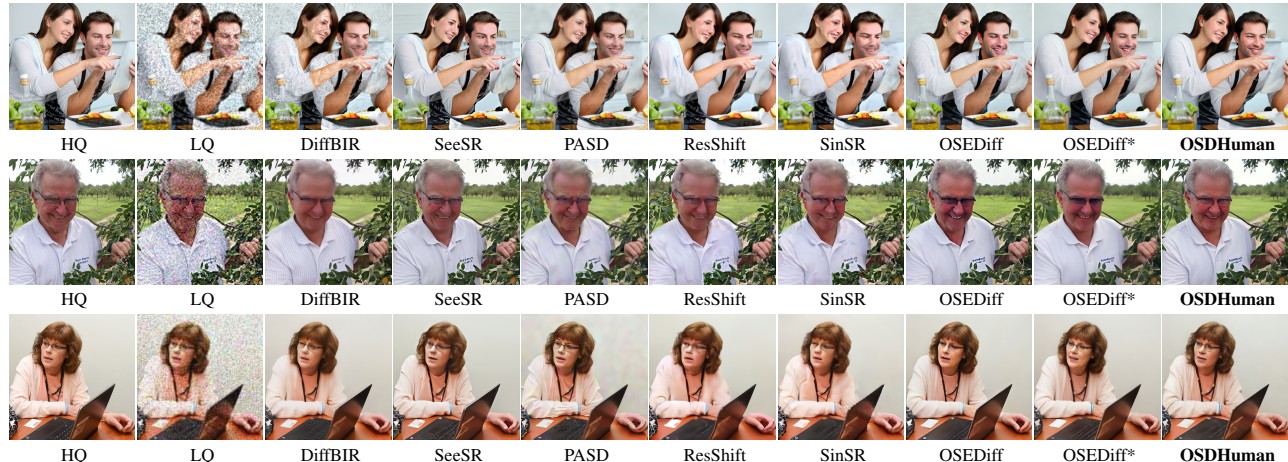

*Figure 9.* Visual comparison of the synthetic PERSONA-Val datasets in challenging cases. Please zoom in for a better view.

| Training Dataset | PERSONA-Val | | | | | | | | | PERSONA-Test | | | |
|---|---|---|---|---|---|---|---|---|---|---|---|---|---|
| | DISTS↓ | LPIPS↓ | PSNR↑ | SSIM↑ | FID↓ | CLIPIQA↑ | MANIQA↑ | MUSIQ↑ | NIQE↓ | CLIPIQA↑ | MANIQA↑ | MUSIQ↑ | NIQE↓ |
| LSDIR | 0.1521 | 0.2692 | **22.51** | 0.6271 | 17.5615 | 0.7229 | 0.6618 | 74.4998 | 3.6461 | 0.6781 | 0.6964 | 73.1266 | 4.6382 |
| **PERSONA** | **0.1414** | **0.2627** | 22.41 | **0.6363** | **16.5987** | **0.7295** | **0.6934** | **76.1256** | **3.5750** | **0.7155** | **0.6977** | **73.7694** | **4.1287** |

*Table 3.* Ablation studies within different training datasets. The best results are highlighted in **bold**.

| Prompt Extractor | | | CLIPIQA↑ | MANIQA↑ | MUSIQ↑ | NIQE↓ |
|---|---|---|---|---|---|---|
| Type | From HQ | From LQ | | | | |
| Null | | | 0.7016 | **0.7226** | 73.1735 | 5.0651 |
| DAPE | ✓ | | 0.6625 | 0.7014 | 72.3104 | 4.9455 |
| HFIE | ✓ | | 0.7111 | 0.6747 | 69.9992 | 5.5031 |
| HFIE | | ✓ | **0.7155** | 0.6977 | **73.76**94 | **4.1287** |

*Table 4.* Ablation studies within different prompt extractors tested on PERSONA-Test. "From HQ / LQ" indicates prompts are extracted from HQ or LQ. The best results are highlighted in **bold**.

color rendering, making it difficult to accurately restore fine details such as facial features. For example, DiffBIR (Lin et al., 2024) suffers from over-smoothing or non-faithful textures, OSEDiff (Wu et al., 2024a) often results in over-saturated facial regions and ResShift (Yue et al., 2023) exhibits distorted facial features. In contrast, our OSDHuman model can restore natural human actions and facial expressions in images, achieving high fidelity and maintaining a high degree of similarity to the original image.

### 4.4. Ablation Studies

**Comparison on Training Datasets.** To evaluate the suitability of our PERSONA dataset for HBR-specific models, we train our OSDHuman on different datasets. The first option employs LSDIR (Li et al., 2023), a dataset widely utilized in the image restoration domain. During training, images in the dataset are randomly cropped to 512×512 as input. The second option employs our PERSONA dataset. As shown in Tab. 3, the results of the model trained on our PERSONA significantly outperform that trained on LSDIR in both PERSONA-Val and PERSONA-Test. This demonstrates that our dataset provides a strong prior for the human body, effectively enhancing the performance of HBR.

**Comparison on Prompt Extractors.** We conduct experiments with four options to evaluate the effectiveness of various prompt extractors for HBR. The first option does not employ a prompt extractor but uses a space as the prompt. For the second option, we use DAPE (Wu et al., 2024b) with the setting of OSEDiff (Wu et al., 2024a). In OSEDiff, HQ images are inputted into DAPE during training to provide higher-quality prompts. Following this, we also input HQ images into HFIE as the third option. The last option, which is our default setting, involves using HFIE to extract prompts from LQ images. As shown in Tab. 4, our HFIE of default setting outperforms the other options. It demonstrates the superior capability of HFIE in providing priors to the model, guiding higher-quality restoration.

### 5. Conclusion

In this work, we propose a high-quality dataset automated cropping and filtering (HQ-ACF) pipeline designed for creating a human body restoration (HBR) dataset to address the lack of a benchmark. Using this pipeline, we develop a person-based restoration with sophisticated objects and natural activities (PERSONA) dataset, which includes training, validation, and test sets. Experimental results show its high quality and suitability for HBR. Additionally, we propose OSDHuman, a one-step diffusion model for HBR. It innovatively employs a high-fidelity image embedder (HFIE) as a prompt extractor. HFIE guides the model in achieving high-quality restoration by effectively extracting and utilizing rich human image features. Extensive experiments show that OSDHuman outperforms current state-of-the-art image restoration methods, which are applied to HBR, in both visual quality and quantitative metrics.

## Impact Statement

This paper presents research aimed at advancing the field of Machine Learning. While there are various potential societal implications of our work, we believe that none of these require particular emphasis here.

## Acknowledgments

This work was supported by Shanghai Municipal Science and Technology Major Project (2021SHZDZX0102) and the Fundamental Research Funds for the Central Universities.

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
