# OpenReview forum: "Human Body Restoration with One-Step Diffusion Model and A New Benchmark"
_ICML.cc/2025/Conference — ICML 2025 poster_

### Official Review · Reviewer_QwNC · 2025-02-24

**Overall Recommendation:** 5

**Summary:**

This paper introduces a high-speed diffusion model that can restore low-quality human body images in just one diffusion timestep. The paper presents a high-quality dataset, PERSONA, which includes diverse human body images. Additionally, the proposed OSDHuman model paves the way for incorporating visual priors into diffusion models for human body restoration. OSDHuman outperforms current state-of-the-art methods in terms of both quality and efficiency on benchmark datasets.

**Claims And Evidence:**

The contributions of this paper are primarily divided into two parts: the PERSONA dataset and the OSDHuman model.

Regarding the dataset, the authors propose a High Quality Human Dataset Pipeline, which uses label filtering, object detection, blur awareness, and IQA filtering to create a dataset of 109,052 high-quality images. The process for constructing this dataset is reasonable, and it would be beneficial for the authors to make it publicly available to contribute to the computer vision and machine learning communities.

Regarding the model OSDHuman, the authors introduce a high-fidelity image embedder (HFIE) and use VSD regularization as guidance. The effectiveness of this approach is demonstrated through Ablation Studies in Table 4. Experimental results demonstrate it perform well in both visual quality and quantitative metrics.

**Essential References Not Discussed:**

The related works cited in the paper are comprehensive.

**Experimental Designs Or Analyses:**

Pros:
In the comparative experiments, the authors retrained the SinSR and OSEDiff models on the PERSONA dataset. The results in Table 2 show that after retraining with PERSONA-train data, the models achieve better performance in human body restoration.

Cons:
From the visual images, it can be seen that the LQ images in PERSONA-Val differ in noise compared to the LQ images in PERSONA-Test. It appears that the Val dataset contains much more severe Gaussian noise than real-world situations. Although the authors mention that the LQ images in the Val dataset are generated using the same degradation pipeline as the training data, could it be more realistic?

**Methods And Evaluation Criteria:**

Pros:
The proposed methods and evaluation criteria are meaningful. The PERSONA dataset fills the gap of lacking publicly available high-resolution open-source datasets for human body images. The proposed model also addresses the gap in portrait photography restoration.

Cons:
However, the images restored from OSDHuman have some color shift. For example, in Figures 5 and 6 in the supplementary materials, the teeth of the person in the 2nd and 5th sets of images are noticeably whiter.

**Other Comments Or Suggestions:**

1. In Table 2, the DISTS column shows that OSEDiff\* performs better than OSEDiff. OSEDiff\* should be cyan.
2. In 2nd row of line 240, `1024` should be `1,024`.

**Other Strengths And Weaknesses:**

Pros:
1. The dataset covers many categories, as shown in Figure 7. It is of high quality and rich in diversity.
2. The efficiency of human body restoration is crucial for applications. The proposed method addresses a practical problem with an efficient solution and favorable qualitative results.
3. Unlike image-to-tag models that generate tags as textual prompts, HFIE directly tokenizes each image, eliminating information loss during the image-tag-embedding process, which can improve fidelity, as mentioned in the supplementary material.
4. The paper is clearly written, easy to follow, and presents a well-motivated and reasonable argument.

Cons:
1. The datasets presented in the article consist of 512x512 size images. Since some of the images do not have the person occupying a large area of the image, does this mean the resolution of these images is too small for the human body? For example, in Figure 6, most individuals occupy less than one-third of the image. Does this imply that the dataset is still not high definition enough?
2. Regarding the model, how scalable is it for images with larger resolutions?

**Questions For Authors:**

1. From Table 2 and Table 4, it seems that using HFIE as a Prompt Extractor does not result in better performance for the MANIQA metric. Could the authors provide an explanation?
2. The authors froze the timestep of the OSD model to 999. I wonder whether this parameter has any effect on the results. Could the authors clarify this?

**Relation To Broader Scientific Literature:**

Human body image restoration has many applications in photography, especially in mobile photography. However, most previous image restoration research has focused on either natural images or faces, such as StableSR[1], SUPIR[2], SinSR[3], DiffBIR[4], and OSEDiff[5]. These models may not perform well for tasks specific on human body images.

As for datasets targeting human bodies, most existing datasets are for fashion purposes [6-7], and there has been a lack of high-resolution portrait datasets for real-world scenes. The dataset proposed in this paper is significant for training more specialized portrait photography image restoration models.

[1] Exploiting Diffusion Prior for Real-World Image Super-Resolution
[2] Scaling Up to Excellence: Practicing Model Scaling for Photo-Realistic Image Restoration In the Wild
[3] SinSR: Diffusion-Based Image Super-Resolution in a Single Step
[4] DiffBIR: Towards Blind Image Restoration with Generative Diffusion Prior
[5] One-Step Effective Diffusion Network for Real-World Image Super-Resolution
[6] Large-scale Fashion (DeepFashion) Database
[7] SCAM! Transferring humans between images with Semantic Cross Attention Modulation

**Theoretical Claims:**

The formulas appear to be correct, with no obvious issues.

---

> ### Author Rebuttal · Authors · 2025-04-01
>
> `Q4-1:` However, the images restored from OSDHuman have some color shift. For example, in Figs. 5 and 6 of the supplementary materials, the teeth of the person in the 2nd and 5th sets of images are noticeably whiter.
>
> `A4-1:`
> Thank you for pointing out the color shift issue. We address this by applying wavelet-based color correction [1], which combines the high-frequency details from the restored image with the low-frequency color components from the original input. This helps align color distributions while preserving structural details.
>
> We plan to integrate this correction into the model for end-to-end learning of color consistency in future work. Your feedback is greatly appreciated.
>
> [1] Mallat, Stephane G. "A theory for multiresolution signal decomposition: the wavelet representation." TPAMI 1989.
>
> ---
> `Q4-2:` The LQ images in PERSONA-Val seem to have stronger Gaussian noise than those in PERSONA-Test. Could the degradation be made more realistic?
>
> `A4-2:`
> In real-world scenarios like group photos, surveillance footage, or chat images, factors such as long shooting distance, compression, and low-end devices often lead to severe degradation. Our pipeline is designed to simulate such challenging cases, which may result in stronger noise than typical natural degradations.
>
> Thank you for the suggestion. We will explore refining the pipeline by incorporating more realistic artifacts like motion blur in future work.
>
> ---
> `Q4-3:` Some individuals in the dataset images occupy less than one-third of the 512×512 frame. Does this mean the resolution is too small for effective human body restoration?
>
> `A4-3:`
> Thanks for raising this concern. Human body restoration is mainly used in portrait photography. In such cases, people care more about the harmony between the person and the background, not just the body or the scene alone. In everyday mobile photography, the human subject does not always take up a large portion of the image. Instead, good composition and visual balance are more important.
>
> The PERSONA dataset uses square 512×512 images. This format works well for various poses, such as standing, sitting, crouching, or group interactions. It also helps blend the human subject with the background, requiring the model to restore both in a consistent and natural way. This is why we chose this design for the dataset.
>
> ---
> `Q4-4:` Regarding the model, how scalable is it for images with larger resolutions?
>
> `A4-4:`
> Thank you for your thoughtful questions. Our model scales well to high-resolution images using a tiled inference strategy. For example, we processed a 3472×4800 image by setting the VAE encoder tile size to 1024×1024 (32px overlap), latent tile size to 96×96 (32px overlap), and decoder tile size to 224×224. Inference ran on an A6000 GPU with a peak memory usage of 32 GB. You can visually inspect the detailed results using the **[anonymous link](https://imgsli.com/MzY0NzIw)**.
>
> ---
> `Q4-5:`
> - In Table 2, the DISTS column shows that OSEDiff* performs better than OSEDiff. OSEDiff* should be marked in cyan.
> - In the second row of line 240, `1024` should be formatted as `1,024`.
>
> `A4-5:`
> Thank you for pointing these out. We will make the necessary corrections.
>
> ---
> `Q4-6:` From Table 2 and Table 4, it seems that using HFIE as a Prompt Extractor does not result in better performance for the MANIQA metric. Could the authors provide an explanation?
>
> `A4-6:`
> Thank you for the observation. MANIQA is a no-reference IQA metric sensitive to the dataset it was trained on. In our paper, we used the PIPAL-trained version, which yielded lower scores for HFIE in Table 4. However, when using the MANIQA model trained on the KonIQ dataset, HFIE achieved the best performance in our ablation study. The results are shown below:
>
> |Type|From HQ|From LQ|MANIQA-PIPAL↑|MANIQA-KonIQ↑|Average↑|
> |-|-|-|-|-|-|
> |Null| | |**0.7226**|0.4430|0.5828|
> |DAPE|✔| |0.7014|0.4309|0.5662|
> |HFIE|✔| |0.6747|0.4718|0.5733|
> |HFIE| |✔|0.6977|**0.4829**|**0.5903**|
>
> This highlights how dataset bias affects MANIQA's judgment. When averaging across both metrics, HFIE performs best, further confirming its effectiveness for human body restoration.
>
> ---
> `Q4-7:` The authors froze the timestep of the OSD model to 999. I wonder whether this parameter has any effect on the results. Could the authors clarify this?
>
> `A4-7:`
> Thank you for your question. In one-step diffusion models, the timestep mainly determines the initial noise level. Since the model performs only a single denoising step, this parameter has limited impact on performance. Fixing it (e.g., to 999) is a common practice, and the model is then fine-tuned to adapt to this noise level.

---

> > ### Comment · Reviewer_QwNC · 2025-04-05
> >
> > I appreciate the answers given and change my score to 5.

---

> > > ### Author Response · Authors · 2025-04-05
> > >
> > > Dear Reviewer QwNC,
> > >
> > > Thank you for your response. We are pleased to learn that our answers addressed your concerns and appreciate your updated score.
> > >
> > > Best regards,
> > > Authors

---

### Official Review · Reviewer_NGq9 · 2025-03-12

**Overall Recommendation:** 3

**Summary:**

The paper proposes a novel approach to human body restoration (HBR) by introducing OSDHuman, a one-step diffusion (OSD) model, and a new benchmark dataset named PERSONA. The authors develop a high-quality dataset automated cropping and filtering (HQ-ACF) pipeline to create PERSONA, which comprises 109,052 high-resolution (512×512) human images covering diverse natural activities and complex interactions. This dataset outperforms existing human-related datasets in quality and richness, addressing the lack of task-specific benchmarks for HBR. OSDHuman incorporates a high-fidelity image embedder (HFIE) to extract precise prompts from low-quality (LQ) images, avoiding misleading guidance, and employs a variational score distillation (VSD) regularizer to align generated outputs with natural image distributions.

**Claims And Evidence:**

Almost correct.

**Essential References Not Discussed:**

N/A

**Experimental Designs Or Analyses:**

Please refer to weaknesses.

**Methods And Evaluation Criteria:**

Almost correct.

**Other Comments Or Suggestions:**

In general, from the perspective of the contribution of the dataset, I think this article is valuable. However, from the perspective of the method, I think it lacks contribution and novelty, as well as detailed theoretical basis. But overall, I think it still has some contribution, so I give it a weak accept now, and I will adjust my score based on the rebuttal.

**Other Strengths And Weaknesses:**

Strengths:

1. The introduction of the PERSONA dataset addresses a critical gap in human body restoration (HBR) research by providing a high-quality, diverse benchmark with 109,052 images, surpassing existing human-related datasets in quality and richness.

2. The paper provides a well-structured experimental evaluation with clear visual comparisons (e.g., Figure 9) and quantitative results (e.g., Tables 2–4), effectively showcasing OSDHuman’s superior visual quality and metric performance on the PERSONA dataset.

Weaknesses:

1. The paper’s innovation is relatively modest, as OSDHuman and the PERSONA dataset build incrementally on existing one-step diffusion techniques and dataset curation methods, offering no transformative advancements in the field of diffusion-based image restoration.

2. The assumption that a single denoising step can effectively restore complex human images lacks rigorous justification, with no formal analysis of the HFIE’s attention mechanism or its convergence properties under severe degradation, potentially impacting the model’s reliability.

3. The article lacks more theoretical analysis of the one-step strategy. Is such a one-step strategy better than multiple steps? Can more theoretical proof be provided?

4. In addition, from a visual comparison, the visual effect of OSEDiff is obviously better than the proposed method, especially it looks more natural. Why is this?

5. While OSDHuman’s performance claims are supported by experimental results, the lack of ablation studies on varied degradation types (e.g., motion blur, compression artifacts) weakens the evidence for the high-fidelity image embedder (HFIE)’s effectiveness.

**Questions For Authors:**

Please refer to weaknesses.

**Relation To Broader Scientific Literature:**

The key contributions of the paper relate to the broader scientific literature by addressing the lack of HBR-specific benchmarks, building on diffusion model advancements, and improving one-step restoration efficiency.

**Theoretical Claims:**

Yes

---

> ### Author Rebuttal · Authors · 2025-04-01
>
> `Q3-1:` The paper's innovation seems limited, as both the model and dataset build on existing work.
>
> `A3-1:`
> Thanks for raising this concern. OSDHuman is the first one-step diffusion model applied to human body image restoration. Compared to traditional multi-step models, it achieves faster inference and lower computational cost, making it practical for real-world scenarios such as group photos or compressed chat images. To reduce bias introduced by external guidance modules, we introduce HFIE, which enables end-to-end training with lightweight and unbiased prompts. For more details on the novelty of HFIE, please refer to `A1-1`.
>
> The PERSONA dataset is also the first large-scale benchmark for this task, addressing prior limitations such as single-person bias, fixed poses, and narrow aspect ratios. It covers diverse poses, interactions, and real-world scenarios and will be open-sourced to support the community.
>
> ---
> `Q3-2:` The assumption that one denoising step can effectively restore complex human images lacks rigorous theoretical justification, and there is no formal analysis of the HFIE’s attention mechanism or its convergence properties under severe degradation.
>
> `A3-2:`
> Thanks for raising this concern. One-step diffusion has been widely explored in image generation and restoration. Recent works [1, 2] distill multi-step models into one step, while methods like SinSR and OSEDiff use one-step strategies for super-resolution. Since body images share similar complexity with natural images, applying one-step diffusion is theoretically reasonable.
>
> The attention mechanism of HFIE can be understood as follows: It encodes 145 embeddings into 77 vectors required by Stable Diffusion 2.1. A learnable query attends to these embeddings via softmax, producing a weighted sum that preserves both local and global information. As a convex combination in the original feature space, this ensures stable and effective guidance during training.
>
> The low-quality images from our degradation pipeline are heavily degraded. OSDHuman with HFIE trains stably on them and achieves good convergence. Compared to DAPE, HFIE leads to faster loss reduction:
>
> |Step|L2 Loss (DAPE → HFIE)|LPIPS (DAPE → HFIE)|
> |-|-|-|
> |10k|0.062 → 0.043|0.843 → 0.771|
> |20k|0.049 → 0.042|0.771 → 0.757|
> |30k|0.047 → 0.041|0.759 → 0.750|
> |40k|0.047 → 0.042|0.753 → 0.753|
>
> Loss visualizations are available at the [anonymous GitHub link](https://anonymous.4open.science/r/Submission_Number-2750-3BA1).
>
> [1] Liu et al., Flow straight and fast: Learning to generate and transfer data with rectified flow, ICLR, 2023.
>
> [2] Yin et al., One-step diffusion with distribution matching distillation, CVPR, 2023.
>
> ---
> `Q3-3:` The article lacks more theoretical analysis of the one-step strategy. Is such a one-step strategy better than multiple steps? Can more theoretical proof be provided?
>
> `A3-3:`
> Thanks for your feedback. One-step diffusion offers a practical trade-off between performance and efficiency, achieving results comparable to multi-step models with much lower latency (see `A2-1`). This is made possible by strong base models with good generalization, which are well-suited for tasks like human body restoration.
>
> As for theoretical reasoning, our work focuses more on introducing a new benchmark and model design rather than formal theoretical proof. We are also eager to see further theoretical development on one-step diffusion models, which would benefit the machine learning community.
>
> ---
> `Q3-4:` In addition, from a visual comparison, the visual effect of OSEDiff is obviously better than the proposed method, especially it looks more natural. Why is this?
>
> `A3-4:`
> Thanks for your questions. In rare cases, our results may look less natural due to specific degradations, sometimes causing color shifts. This can be mitigated via post-processing like wavelet-based correction [3], as described in more detail in `A4-1`. We will continue to refine our model to improve visual consistency, especially in challenging scenarios.
>
> Overall, our method outperforms OSEDiff in preserving fine facial details and natural tones. For example, in Fig. 2, expressions like subtle smiles are better retained, while OSEDiff may distort them and introduce unnatural reddish hues, indicating weaker perceptual consistency.
>
> [3] Mallat, Stephane G., A theory for multiresolution signal decomposition: the wavelet representation, TPAMI, 1989.
>
> ---
> `Q3-5:` The lack of ablations on varied degradations (e.g., motion blur, compression) weakens the evidence for HFIE’s effectiveness.
>
> `A3-5:` The degradation model we used, Real-ESRGAN, includes several common degradation types, such as downsampling, noise, blur, and JPEG compression. We appreciate your suggestions, and we plan to explore additional degradation types to guide the model to perform better in more natural scenarios. We will experiment with motion blur and continue to explore the model’s effectiveness across a wider range of scenarios.

---

### Official Review · Reviewer_pp6W · 2025-03-13

**Overall Recommendation:** 4

**Summary:**

This paper presents a dataset automated cropping and filtering pipeline and constructs a person-based restoration with sophisticated objects and natural activities dataset. A novel one-step diffusion model is proposed for human restoration. Experimental results demonstrate the effectiveness.

**Claims And Evidence:**

Yes

**Essential References Not Discussed:**

Essential references are discussed.

**Experimental Designs Or Analyses:**

The experimental design and analysis is sound.

**Methods And Evaluation Criteria:**

Yes

**Other Comments Or Suggestions:**

1. Comparisons with non-diffusion restoration models could be added.

**Other Strengths And Weaknesses:**

Pros:

1. This work is well organized and well written.

2. A new dataset is proposed which is of research value.

3. Experimental presentations are extensive.


Cons:

1. It is necessary to show the inference speed of different algorithms. This is the reason why one-step diffusion models are used.

2. In ablation experiments, the proposed components did not always improve on all metrics. It is recommended to add visual comparisons to show the effectiveness of the proposed components.

3. The aim of this paper is to present the dataset and the corresponding methodology for human body restoration. However, the comparison of the visualizations in Figures 8, 9 is still focusing on the face region and the gaps in other human body regions are not significant. I am concerned about the value of this study and how it differs from face restoration.

**Questions For Authors:**

Please see weakness.

**Relation To Broader Scientific Literature:**

Related to the diffusion models and blind image restoration literature.

**Theoretical Claims:**

No theoretical proofs involved.

---

> ### Author Rebuttal · Authors · 2025-04-01
>
> `Q2-1`: It is necessary to show the inference speed of different algorithms. This is the reason why one-step diffusion models are used.
>
> `A2-1:`
> Thank you for your thoughtful suggestion. We have provided a detailed comparison of inference speed, parameter count, and computational cost for several recent diffusion models in the supplementary materials. For your convenience, the table below presents the same results:
> | Methods| DiffBIR | SeeSR | PASD | ResShift | SinSR | OSEDiff | **OSDHuman (Ours)** |
> |----|---------|-------|------|----------|-------|---------|-------|
> | Step | 50 | 50 | 20 | 15 | 1 | 1 | 1 |
> | Time (s) ↓| 9.03 | 5.05| 3.15 | 2.88| 0.19| 0.13| **0.11** |
> | Param (M) ↓| 1717 | 2524 | 1900 | 119 | 119 | 1775 | 1576 |
> | MACs (G) ↓| 24234 | 65857 | 29125 | 5491 | 2649 | 2265 | **2200** |
>
> ---
> `Q2-2:` In ablation experiments, the proposed components did not always improve on all metrics. It is recommended to add visual comparisons to show the effectiveness of the proposed components.
>
> `A2-2:`
> Thank you for your suggestion. Regarding the observation that the proposed component (HFIE) did not always improve on all metrics in the ablation study (Table 4), we have discussed this in detail in our response to `Q4-7`. We believe that HFIE is not inferior to other methods under the MANIQA metric.
>
> We appreciate your suggestion and will include additional visual comparisons to illustrate the effectiveness of each component. You can view the visual comparisons at the [anonymous GitHub link](https://anonymous.4open.science/r/Submission_Number-2750-3BA1).
>
> ---
> `Q2-3:` This paper aims to present the dataset and the corresponding methodology for human body restoration. However, the comparison of the visualizations in Figs. 8 and 9 is still focusing on the face region and the gaps in other human body regions are not significant. I am concerned about the value of this study and how it differs from face restoration.
>
> `A2-3:`
> Thanks for raising this concern. Firstly, the reason we focus on faces in Figs. 8 and 9 emphasizes our method's effectiveness in restoring small-scale facial details. Since humans are particularly sensitive to facial perception, improvements in facial regions are noteworthy. Secondly, our method does not solely focus on faces; it also achieves significant restoration results for other body regions and background areas. Additional visual comparisons demonstrating these broader improvements can be found in the supplementary materials.
>
> ---
> `Q2-4:` Comparisons with non-diffusion restoration models could be added.
>
> `A2-4:`
> Thank you for your suggestion. We will add additional comparisons with classic non-diffusion restoration models. The test set comparison results are shown in the table below:
> | Methods| CLIPIQA↑| MANIQA↑| MUSIQ↑| NIQE↓|
> |---|----|---|---|---|
> | Real-ESRGAN[1]| 0.4721 | 0.6159| 67.7610 | 4.4390 |
> | BSRGAN[2] | 0.5307 | 0.6159 | 70.8345 | 4.4474 |
> | SwinIR[3] | 0.4847 | 0.6240 | 69.7636 | **4.0014** |
> | DAT[4] | 0.3194 | 0.3497 | 27.8132 | 8.3351 |
> | HAT[5] | 0.3936 | 0.5563 | 52.5818 | 6.4262 |
> | **OSDHuman (Ours)** | **0.7155** | **0.6977** | **73.7694** | 4.1287 |
>
> These results demonstrate the effectiveness of our proposed method compared to existing non-diffusion approaches.
>
> [1] Wang et al., Real-ESRGAN: Blind super-resolution with pure synthetic data, ICCV, 2021
>
> [2] Zhang et al., Designing a practical degradation model for deep blind image super-resolution, ICCV, 2021
>
> [3] Liang et al., SwinIR: Image restoration using Swin transformer, ICCVW, 2021
>
> [4] Chen et al., Dual aggregation transformer for image super-resolution, ICCV, 2023
>
> [5] Chen et al., Activating more pixels in image super-resolution transformer, CVPR, 2023

---

### Official Review · Reviewer_ndyH · 2025-03-13

**Overall Recommendation:** 3

**Summary:**

This study addresses the challenge of human body restoration by introducing a high-quality dataset construction pipeline, HQ-ACF, which automatically crops and filters human images from existing datasets. Using this pipeline, the PERSONA dataset is created, offering superior quality and content richness compared to existing human-related datasets. Additionally, the study proposes OSDHuman, a novel one-step diffusion model for human body restoration. OSDHuman features a high-fidelity image embedder (HFIE) to generate more accurate prompts, reducing the risk of misleading guidance. Experimental results demonstrate that OSDHuman achieves state-of-the-art performance in both visual quality and quantitative metrics.

**Claims And Evidence:**

Yes.

**Essential References Not Discussed:**

The authors have cited the related works that are essential to understanding the key contributions.

**Experimental Designs Or Analyses:**

Yes.

**Methods And Evaluation Criteria:**

Yes.

**Other Comments Or Suggestions:**

There are some minor issues that need to be addressed:

1. The full name of SOTA is missing in Line 107.
2. The subcaption of Figure 2 should be "SinSR" instead of "Sinsr."
3. Some citation formats should be revised. For example, "(Liu et al., 2021a) represent the texture details of the human body using ..."
4. Mathematical expressions should be consistent: "z_L" is used in Line 226, while "Z_L" appears in Figure 4.
5. LoRA is not cited.

**Other Strengths And Weaknesses:**

#### **Strengths**

1. The effort in constructing a large-scale dataset specifically for human body restoration is commendable. The proposed HQ-ACF pipeline effectively leverages existing datasets to curate high-quality human images, addressing the scarcity of benchmark datasets in this domain.

2. The proposed OSDHuman model achieves state-of-the-art performance on the newly introduced PERSONA dataset, demonstrating its effectiveness in restoring human images with improved visual quality and quantitative metrics.

#### **Weaknesses**

1. The novelty of the proposed approach is somewhat limited. The concept of the VSD is directly adapted from a previous work [1], and the HFIE could be seen as an attention-based variant of the DAPE framework. More justification and discussion on the unique contributions of the method would strengthen the paper.

2. The degradation types applied in the dataset are limited and not thoroughly discussed. If the dataset primarily uses blind super-resolution from [2] as its degradation process, it would be more accurate to frame the problem as "human body super-resolution" rather than the broader term "human body restoration." A broader range of degradations would enhance the dataset’s applicability.

3. The superiority of the dataset is claimed based on its improved IQA values, yet the dataset construction process involves discarding images with lower IQA scores. This raises concerns about potential bias in evaluation and weakens the contribution of the HQ-ACF pipeline. A more transparent discussion on dataset selection criteria and its impact on evaluation would be beneficial.

4. The test set is entirely sourced from the VOC dataset, while the training set is compiled from multiple datasets. This discrepancy could lead to biased evaluations, as the test set may not fully represent the diversity of the training data. A more diverse and representative test set would provide a better assessment of model generalization.

5. The evaluation is limited to the newly introduced PERSONA dataset, without testing on existing human restoration or super-resolution datasets. Assessing the model’s performance on established datasets would better demonstrate its generalizability and highlight its advantages over prior methods.

[1] Wu, R., Sun, L., Ma, Z., and Zhang, L. One-step effective diffusion network for real-world image super-resolution. In NeurIPS, 2024a.

[2] Wang, X., Xie, L., Dong, C., and Shan, Y. Real-esrgan: Training real-world blind super-resolution with pure synthetic data. In ICCV, 2021b.

**Questions For Authors:**

Please refer to Weakness.

**Relation To Broader Scientific Literature:**

Some of the key contributions of the paper are from the previous works such as [1].

[1] Wu, R., Sun, L., Ma, Z., and Zhang, L. One-step effective diffusion network for real-world image super-resolution. In NeurIPS, 2024a.

**Theoretical Claims:**

NA, there is not Theoretical Claim.

---

> ### Author Rebuttal · Authors · 2025-04-01
>
> `Q1-1:` The novelty seems limited, as VSD is adapted from prior work and HFIE resembles an attention-based DAPE.
>
> `A1-1:`
> Thank you for your valuable comments. Our model is the first to focus on human body restoration using a one-step diffusion framework. Our model's VSD module follows the OSEDiff design, which builds on DMD [2] to optimize distribution loss in the latent state space. The DAPE, which requires additional training, inevitably introduces errors and missed predictions, as discussed in our supplementary materials. In contrast, our proposed HFIE does not require separate training. Instead, HFIE utilizes the image encoder from the Recognize Anything Model and integrates a trainable multi-head attention layer. This approach improves performance and reduces computational costs by eliminating the need for extra training and tagging heads.
>
> [1] Wu et al., One-step effective diffusion network for real-world image super-resolution, NeurIPS, 2024
>
> [2] Yin et al., One-step diffusion with distribution matching distillation, CVPR, 2023
>
> ---
> `Q1-2:` The dataset uses limited degradation types. If it mainly relies on blind SR from [4], should the task be framed as "super-resolution" rather than "restoration"?
>
> `A1-2:`
> Thanks for raising this concern. The RealESRGAN [4] pipeline models a broad range of realistic degradations, including blur, noise, and JPEG compression, beyond simple downsampling. After applying this pipeline, we resample images to 512×512, preserving diverse artifacts.
>
> In addition to synthetic validation data, our test set includes real-world degraded images with motion blur, noise, and compression. Examples are shown in Figs. 5 and 6 of our supplementary materials.
>
> [4] Wang et al., Real-ESRGAN: Training real-world blind super-resolution with pure synthetic data, ICCV 2021
>
> ---
> `Q1-3:` Since low-IQA images were filtered out, does this introduce evaluation bias and weaken the value of the HQ-ACF pipeline?
>
> `A1-3:`
> Thanks for raising this concern. The IQA metrics used in our HQ-ACF pipeline are widely recognized and commonly applied in image restoration tasks, ensuring the high quality of our dataset. To address potential bias, we further evaluated the dataset using additional IQA metrics not involved in the selection and compared it with other human-related datasets. As shown in the table below, PERSONA achieves the best overall quality and the highest RAM++ category diversity, indicating not only consistently high data quality but also richer semantics.
>
> |Dataset|BRISQUE[5]↓|HyperIQA[6]↑|TOPIQ_NR[7]↑|LIQE[8]↑|RAM++ Categories↑|
> |-|-|-|-|-|-|
> |VOC|21.28|0.608|0.608|4.241|2759|
> |iDesigner|25.80|0.632|0.647|4.388|1167|
> |DeepFashion|42.19|0.639|0.650|4.681|2496|
> |CrowdHuman|20.43|0.531|0.525|3.690|2220|
> |**PERSONA (Ours)**|**10.38**|**0.652**|**0.661**|**4.878**|**3365**|
>
> [5] Mittal et al., Blind/referenceless image spatial quality evaluator, Asilomar Conference on Signals, Systems, and Computers, 2011
>
> [6] Su et al., Blindly assess image quality in the wild guided by a self-adaptive hyper network, CVPR, 2020
>
> [7] Chen et al., TOPIQ: A top-down approach from semantics to distortions for image quality assessment, IEEE TIP, 2024
>
> [8] Zhang et al., Blind image quality assessment via vision-language correspondence: A multitask learning perspective, CVPR, 2023
>
> ---
> `Q1-4:` Since the test set only uses VOC, does it fully reflect the diversity of the training data?
>
> `A1-4:`
> The VOC dataset contains images that generally exhibit more severe degradations compared to more recent datasets. Thus, we consider it suitable for evaluating the model's restoration capabilities. We appreciate the suggestion and will include more diverse sources in the public release to enhance test set representativeness.
>
> ---
> `Q1-5:` Evaluation is only on PERSONA. Would testing on existing datasets better demonstrate generalizability?
>
> `A1-5:`
> Thank you for your thoughtful questions. Our proposed PERSONA dataset is the first publicly available benchmark specifically designed for human body restoration. Previous human body restoration methods [5, 6] have not provided public access to their test datasets. We hope that the release of the PERSONA dataset and benchmark will facilitate further contributions to the machine learning and computer vision communities.
>
> [5] Zhang et al., Diffbody: Human body restoration by imagining with generative diffusion prior, arXiv:2404.03642, 2024
>
> [6] Wang et al., Prior based pyramid residual clique network for human body image super-resolution, Pattern Recognition, 2024
>
> ---
> `Q1-6:` There are some minor issues that need to be addressed:
> 1. The full name of SOTA is missing in Line 107.
> 2. The subcaption of Fig. 2 should be "SinSR" instead of "Sinsr."
> 3. Some citation formats should be revised.
> 4. Mathematical expressions should be consistent: "z_L" is used in Line 226, while "Z_L" appears in Fig. 4.
> 5. LoRA is not cited.
>
> `A1-6:`
> Thank you for pointing out these issues. We'll carefully revise them.

---

### Decision · Program_Chairs · 2025-05-01

**Decision:**

Accept (poster)

**Comment:**

The paper received all accept recommendations. AC agrees with this recommendation and therefore is happy to accept the paper. Please address reviewer comments to the camera-ready version of the paper.